

# On the Use of Routine Airborne Observations for Evaluation and Monitoring of Satellite Observations of Thermodynamic Profiles

Timothy J. Wagner[1], Thomas August[2], Tim Hultberg[2] and Ralph A. Petersen[1]

[1]Cooperative Institute for Meteorological Satellite Studies (CIMSS), Space Science and Engineering Center (SSEC), University of Wisconsin – Madison, Madison, Wisconsin, 53708, United States of America
[2]EUMETSAT, Darmstadt, 64295, Germany

*Correspondence to*: Timothy J. Wagner (tim.wagner@ssec.wisc.edu)

**Abstract.** Satellite-based observations require independent sources of data to monitor and evaluate their precision and accuracy. For the temperature and water vapor profiles produced by satellite-based sounders, this often results in comparisons to operational radiosondes. However, polar-orbiting satellite overpasses are frequently misaligned with the global synoptic launch times. The routine airborne in situ observations of temperature and water vapor from the Airborne Meteorological Data Relay (AMDAR) program and Water Vapor Sensing System-II (WVSS-II) instrument greatly enhance opportunities for making precise matchups due to the far greater temporal frequency and spatial density of aircraft flights.

The potential for the use of aircraft-based observations as a source for evaluation of tropospheric satellite sounder profiles is explored through a year-long intercomparison with the IASI Level 2 profiles produced from both the Metop-A and Metop-B satellites. Results using 1 h and 50 km match criteria indicate good agreement between the satellites and the aircraft-based observations with temperature, specific humidity, and relative humidity biases generally less than 0.5 K, 0.8 g kg-1, and 5 % respectively; both IASI instruments perform nearly identically. While the intercomparisons are generally limited to the troposphere as aircraft typically reach their maximum height at the tropopause, the substantially larger number of intercomparison points enable characterization as a function of season, scan angle, and other characteristics heretofore unexplored due to a lack of validation data.

## 1 Introduction

The advantages of low-earth orbiting satellites (LEO, also known as polar-orbiting satellites) are well known, such as global coverage with coordinated sets of high spatial resolution instruments and frequent observations of polar regions. The impact of LEO observations on both atmospheric research and operational meteorology has been felt for over sixty years, as products obtained from these systems have progressed from simple black-and-white visible wavelength snapshots of the location and extent of daytime cloud clover to well-calibrated quantitative measurements of atmospheric and surface properties. One particularly useful application of satellite remote sensing of the atmosphere is the retrieval of thermodynamic profiles. Through passive remote sensing of upward atmospheric emission in the infrared and/or microwave bands and judicious use of statistical or physical retrievals, it is possible to obtain accurate vertical profiles of temperature and water vapor in the atmosphere.



Information about atmospheric structure and stability is now available in places where such observations are otherwise sparse, such as over oceans, polar regions, and less-developed land areas. Thermodynamic soundings from LEO have a diverse set of applications, including weather forecasting and nowcasting, climate monitoring, (Schröder et al., 2018), and atmospheric

composition (Clerbaux et al., 2009) where they are inputs to subsequent geophysical retrievals. In particular, the improvements made over the last years with the precision and latency of the retrievals have enabled these retrievals to become important assets to further support the work of forecasters (Smith et al., 2021; Bloch et al., 2019; Herold and Hungershofer, 2019; Kocsis et al. 2018). Knowledge developed in support of current polar missions are also preparing the groundwork for nowcasting applications of the upcoming geostationary InfraRed Sounder (IRS) onboard the Meteosat Third Generation (MTG, Holmlund

et al., 2021). MTG-IRS will have the decisive advantage of an unprecedented three-dimensional look into the atmosphere with a substantial improvement in temporal resolution: every 30 min over Europe compared to the twice-daily revisits offered by a polar sounding.

For successful scientific and operational applications, it is essential to monitor the performances of the satellite products throughout the mission lifetime. Traditionally, the standard for evaluating thermodynamic profiles remains the balloon-borne

radiosonde due to its ubiquity and well-characterized precision. However, the utility of the operational radiosonde network in evaluating LEO products is limited due to the large spatial and temporal gaps (up to several hours) between radiosonde launches and satellite overpasses. Whereas radiosonde observations are usually launched at standard synoptic observation times (such as 0000 and 1200 UTC), LEO overpasses are often synchronized with the sun so that insolation and solar zenith angles can be relatively constant for all daytime observations over the course of a day. This does mean, however, that

performance can only be monitored at certain locations in which radiosonde launches and LEOS overpasses coincide within an acceptable time frame. For example, at EUMETSAT the operational comparison window is up to 3 h between radiosonde and satellite observations.

One possibility for augmenting the radiosonde validation matchups for LEO thermodynamic products is the in situ observations from the commercial aviation network. Modern jetliners need to constantly monitor the atmospheric state (including winds,

pressure, and temperature) to safely operate and navigate, and these observations have significant value to both the research community as well as national weather services. The World Meteorological Organization (WMO) has developed the Atmospheric Meteorological Data Relay (AMDAR; Moninger et al. 2003) program to collect, control for quality, and disseminate these observations in near-real time. With every take-off and landing, aircraft from AMDAR-participating profile the depth of the troposphere, and while cruising between airports these aircraft report substantial information about the near-

tropopause environment and lower stratosphere. In addition, approximately 100 aircraft operating in the United States have



also been equipped with the Water Vapor Sensing System – II (WVSS-II, Petersen et al., 2016) to measure specific humidity (SH).

In the present work, we studied the use of AMDAR and WVSS-II observations for the validation of atmospheric temperature and humidity profiles retrieved from satellite sounders. The methodology was developed to evaluate and characterize the performance of the Infrared Atmospheric Sounding Interferometer (IASI) (Cayla, 1993) Level 2 (L2) temperature and water vapor products from the EUMETSAT Polar System (EPS; Klaes et al., 2007, 2021) over the continental United States (CONUS). However, this same technique can be applied to other similar satellite sounder products including those originating from geostationary orbits. We have chosen to conduct this study over CONUS as it offers a diverse set of meteorological conditions and surface types coupled with a high density of airborne observations (especially water vapor observations) relative to the rest of the planet. The large number of AMDAR observations fosters many more intercomparison points than is possible with radiosondes alone, enabling new categories of intercomparisons that would otherwise be difficult to perform with other datasets. The remainder of this paper explores the performance of IASI relative to AMDAR observations in a variety of different categorizations and offers insight into how the airborne observations can serve as part of an operational evaluation system for any satellite profiling system.

## 2. Instrumentation

### 2.1 IASI observations and level 2 products

The EUMETSAT Polar System (EPS; Klaes et al., 2007; Klaes et al., 2021) consists of three satellites, Metop-A, -B, and -C, launched in 2006, 2012 and 2018, respectively. These satellites have a sun-synchronous orbit at a mean altitude of 817 km and a period of 101 minutes with a 29-day ground track repeat cycle. One of the primary instruments is IASI (Chalon et al., 2001; Blumstein et al., 2004; Hilton et al., 2010), a hyperspectral Fourier transform interferometer that observes over the spectral range of 640 to 2700 cm$^{-1}$ (3.6 to 15.5 µm) with a spectral sampling of 0.25 cm$^{-1}$ (0.5 cm$^{-1}$ resolution), a horizontal resolution at nadir of 12 km, and a swath width of approximately 2000 km which enables global coverage twice a day. The retrieval of geophysical parameters also exploits observations from the microwave companion instruments: the Advanced Microwave Sounding Unit (AMSU) and the Microwave Humidity Sounder (MHS). This infrared-microwave synergy allows for complete vertical profiling including in cloudy environments. The combined retrieval is referred to as IASI Level 2 products out of convenience. IASI-only retrievals are the fallback mode in case the microwave sensors are unavailable, but evaluating the performance of this mode is out of scope of this study.

The retrieval methodology (Hultberg and August, 2014) is based on machine learning techniques and constitutes the operational baseline of all EUMETSAT hyperspectral missions, both current (IASI) and future [Meteosat Third Generation Infrared Sounder (MTG–IRS), IASI–Next Generation]. It implements a piecewise linear regression, or PWLR. In this



approach, a training base is first constituted with of the order of $10^8$ of real IASI and AMSU/MHS observations collocated with model reanalysis data from ECMWF (ERA-5; Hersbach et al., 2020). The observed spectra (predictors) and the atmospheric profiles (predictands, originally on 137 surface-dependent pressure levels) are represented in principal

components scores (PCS). The satellite observations, once in PCS, are partitioned in observation classes by application of k-mean clustering (MacQueen, 1967). A linear regression is then performed in each individual observations class between the IASI+AMSU/MHS observations and the geophysical parameters from ERA-5. Once the regression coefficients are derived, a second linear regression is applied between the observations (in PCS) and the absolute training error in the lower troposphere (i.e. the difference between the training and retrieved quantities at the bottom of the troposphere). This forms an uncertainty

estimate which users can use as quality indicator to perform data selection adapted to their applications.

In the retrieval stage, the satellite measurements are mapped onto the observation classes and the regression coefficients are applied to retrieve the geophysical information as well as the uncertainty estimates. Further details are provided in the MTG-IRS L2 Algorithm Theoretical Baseline Document (EUMETSAT, 2021). Longer-term records of temperature and humidity

products like those evaluated here have been subject to numerous radiosonde-based validation studies, essentially using radiosondes (EUMETSAT 2022; EUMETSAT, 2018; EUMETSAT, 2016; Feltz et al., 2017; Roman et al., 2016; Boylan et al., 2015). Regional products are available with a latency of 15 to 30 min and global products are available within 1.5 h.

As sun-synchronous polar orbiting satellites, Metop-A and Metop-B have an ascending node covering the CONUS at night

(roughly 9:00 to 10:00 PM local time depending on location and time zone) and a descending node during the day (again, roughly 9:00 to 10:00 AM).   These are both periods in which substantial aircraft reports are available over much of the CONUS. With radiosonde launches timed to observe at 0000 and 1200 UTC, this means that CONUS-based radiosondes are valid for local times between 4:00 and 8:00 AM and PM.  While eastern radiosonde launches tend to align with Metop overpasses, the western launches are well before Metop arrives.  This limits the potential satellite/radiosonde intercomparisons

to specific regions and illustrates why additional sources of evaluation data may be desired.

## 2.2 AMDAR observations

With so many high-quality in situ observations being made by commercial aircraft every day, it is natural that the aviation industry and the meteorological community have partnered together to exploit their benefits. Through the AMDAR program, participating airlines share their meteorological data with each other and with various national weather agencies. Costs for the

system, which mostly consist of data transfer and quality control, are generally borne by the weather agencies and are substantially less than the operational costs of even a modest network of consumable radiosondes (WMO, 2014).

The observations are transmitted to the surface via the different air-to-ground communications protocols in place throughout the world. In North America, this is accomplished via the Aircraft Communications Addressing and Reporting System





(ACARS), which is another name by which the airborne observations are sometimes known. Participation in the AMDAR program is voluntary, but most major carriers in the United States and western Europe contribute observations. Data coverage is densest over those regions, which means that the locations of the observations tend to be biased toward more populated regions of the Northern Hemisphere. Several studies evaluating the accuracy of AMDAR observations have been carried out; Zhang et al., (2018) summarizes many of them. More recently, Wagner and Petersen (2021) performed a year-long CONUS-

wide evaluation of AMDAR-observed temperatures against operational radiosondes and found excellent agreement between the two systems, with a small cool bias of 0.2 K with a standard deviation of 0.8 K. It is unsurprising, therefore, that with so many well-characterized measurements, assimilating AMDAR observations into numerical weather prediction (NWP) models has been found by numerous studies to have a significant positive impact on forecasts; a review of many of these studies can be found in Petersen (2016).


There are some differences between AMDAR observations and radiosondes that are worth noting that largely arise out the airlines' primary purpose of the safe transfer of people and goods instead of meteorological data collection. Regular diurnal, hebdomadal, and annual cycles in the number of observations correspond with typical fluctuations in air traffic (e. g. more during the day than at night, more on weekdays than weekends, and more during winter holidays and summer than other

months). Few observations are made within or adjacent to severe storms or tropical systems, profiles rarely extend above 160 hPa as most planes do not cruise that high, and large-scale disruptions to air travel like the COVID-19 pandemic can greatly reduce the number of observations. Despite these limitations, the AMDAR dataset provides far greater spatial and temporal density throughout the troposphere than is possible with the radiosonde network.

## 2.3 Water Vapor Sensing System-II (WVSS-II) observations

Water vapor observations do not have the same immediate utility to aircraft in flight that observations of temperature, pressure, and winds do. Therefore, humidity sensors are not included as standard equipment by the major aircraft manufacturers. However, the atmospheric science community has recognized that significant value would be added to the already beneficial AMDAR observations if they could be augmented with in situ measurements of water vapor. This has culminated in the creation of the WVSS-II sensor, a laser diode that directly counts individual water vapor molecules; from that, the system is

able to calculate SH with a high degree of accuracy. In the United States, certain Boeing 737 airplanes from Southwest Airlines and Boeing 757 airplanes operated by UPS Airlines have been equipped with WVSS-II. These two carriers complement each other well, since the passenger carrier Southwest tends to operate during daytime and early evening while UPS, as a freight carrier, usually operates during overnight hours to facilitate next-day shipping. Additional WVSS-II observations are available in Europe from Airbus A320 planes serving short-haul destinations out of Lufthansa's Frankfurt, Germany, hub. Williams et

al. (2021) and Wagner and Petersen (2021) conducted CONUS-wide comparisons between WVSS-II and operational National Weather Service radiosondes and found good agreement between the two systems, with a bias of approximately 0.3 g kg$^{-1}$ and a standard deviation of approximately 1 g kg$^{-1}$ near the surface that decreases with height as absolute water vapor content





decreases. The WVSS-II observations have also been shown to have a significant positive impact on NWP (Hoover et al.,
2017; Petersen et al., 2016). A map showing the distribution of WVSS-II observations across CONUS is shown in Fig. 1
(bottom).  For the sake of convenience, this paper uses the term AMDAR to include all the aircraft-based observations, whether
or not water vapor observations are included.

## 3. Methodology

The present work uses the entirely of IASI and AMDAR observations for the calendar year of 2017. This represents a period
in which two separate IASI-supporting satellites, Metop-A and Metop-B, were operational and the significant flight disruptions
due to the Covid-19 pandemic had yet to be realized. Observations were considered to be matched if they occurred within 50
km and +/- 1 h.  Since this matching radius is larger than the footprint of an IASI pixel, the same AMDAR observation could
be matched to more than one IASI pixel simultaneously while multiple AMDAR observations could be matched to the same
IASI observation. IASI profiles were interpolated onto a vertical grid with three bins per 100 hPa of height, and the
observational differences were calculated between AMDAR observations in pressure altitude, which is easily converted to
pressure using the standard atmosphere. AMDAR profiles were also interpolated to that vertical grid to facilitate the
intercomparisons. Since the WVSS-II measures SH, these observations were also combined with coincident temperature
reports to derive RH at each reported pressure level. The quality control applied here retained Metop retrievals where the
uncertainty estimates on temperature and dew point profiles are better than 1.5 K and 2.5  respectively.


It is important to note that this study accounts for the spatial drift of the observations when doing the matching; since an
airplane usually undergoes significant horizontal displacement during ascent, a given airplane may be matched to one IASI
profile near the surface and a different profile after having ascended or descended for a period. Many previous IASI validations,
including the operational comparisons carried out at EUMETSAT, have used a no-drift assumption with respect to the
radiosonde as many operational radiosonde data feeds do not retain geographical coordinates beyond the launch site. Each
AMDAR observation includes the latitude and longitude of the observation which makes these direct geographic comparisons
possible.

Throughout this paper, the bias is calculated as the mean of IASI minus AMDAR differences at a specific height, meaning that
the aircraft-based observations are the reference state for the intercomparison. The Williams et al. (2021) and Wagner and
Petersen (2021) studies showed that the airborne observations compare very favourably with operational National Weather
Service (NWS) radiosondes over similar spatial and temporal domains, which makes this assumption appropriate. The standard
deviations of the differences were calculated as well, and these results are shown throughout Section 4.





## 4. Intercomparisons

### 4.1 Global statistics for Metop-A and Metop-B separately

Since the dataset consists of two different IASI instruments, the first step in the analysis is to determine if the separate instruments exhibit similar behaviours. Results from this investigation are shown in Fig. 2. Overall, the two instruments agree remarkably well, with effectively identical biases and standard deviations at all analyzed heights. The overall pattern of IASI performance relative to AMDAR can also be discerned from Fig. 2. Except at the surface where IASI is effectively unbiased, the IASI temperature retrievals (Fig. 2 left) have a slight cool bias of 0.2 to 0.4 K up to 260 hPa. Above that height, the magnitude of the cool bias increases to approximately 0.6 K at 200 hPa. The standard deviation of the differences in the temperatures, which is a measure of uncertainty in the retrieval, is at its greatest at the surface with a value of approximately 2.4 K. This decreases with height to be only about 1 K in the middle to upper troposphere (400 to 500 hPa), at which point it increases again to be 2 K at 200 hPa. This largely aligns with the profile of bias and standard deviation of differences between IASI and radiosondes, although the magnitude of the standard deviations is larger for the AMDAR comparisons than it is for the radiosonde ones (EUMETSAT, 2022). The SH bias (Fig. 2 centre) is dry and greatest at the surface at approximately 0.8 g kg$^{-1}$ and decreases with increasing altitude to near zero at 200 hPa. Since SH is an absolute measure of water vapor content, which decreases with height due to decreasing temperature and increasing distance from evaporative sources, this decrease is expected. Likewise, the standard deviations in water vapor differences also decrease with height, from 2.2 g kg$^{-1}$ to about 0.4 g kg$^{-1}$ at 200 hPa. Dry biases in humidity have not necessarily observed to that extent when evaluated with respect to radiosondes (EUMETSAT, 2022), however it is important to note that the geographical and diurnal sampling are different; the eastern and southern United States contains the most water vapor intercomparisons (Fig. 1(b)) which has moisture content that might not be consistent with the global regions sampled by the operational radiosonde intercomparisons. Previous work has shown a small moist bias in AMDAR observations when compared to radiosondes of the same order (Wagner and Petersen, 2021) which could account for part of the bias reported here. The random component of the differences (precision) in SH is consistent with the results from radiosonde intercomparisons.

Because the end-user requirements for IASI were also specified in terms of RH, it is useful to examine the comparisons for that measure (Fig. 2, right). The differences will reflect the combined effects of biases already quantified in temperature and water vapor. In this case, the RH bias is consistently between 2 % and 4 % from the surface to 250 hPa at which point it steadily increases to 8 % at 200 hPa. It is interesting to note that the respective changes in the offsetting temperature and water vapor biases with height in the lower troposphere nearly balance each other out, producing a RH bias profile that is nearly constant with height. Near the tropopause, the SH levels are so low that the temperature bias dominates the RH resulting in nearly identical curve shapes in both Fig. 2 (left) and Fig. 2 (right) above 250 hPa. The standard deviation in RH fluctuates around 15 % which, factoring in the intrinsic AMDAR uncertainties as well as the horizontal and vertical collocation and representativeness uncertainties, is consistent with the user requirements for IASI of 10 % precision in RH on 1 to 2 km layers.





Iit is important to note that the differences in Fig.2, while very small, are statistically significant: at certain heights the bias

between Metop-A and Metop-B differs by less than 0.01 K, yet according to a two sample t-test that difference is still statistically significant at the 95% confidence interval due to the thousands of observations present at that height. In a practical sense, however, the two datasets are functionally identical with changes in bias that are well within the uncertainty of the instruments themselves.  Because of this, subsequent analyses will focus on a single large combined dataset comprising of both Metop-A and Metop-B. The significant spatial and temporal density of AMDAR observations provides opportunity to

evaluate satellite performance using previously unassessed matchup characteristics.  Several examples of questions that cannot be addressed using traditional twice daily synoptic radiosonde observations as a ground truth for validating LEO retrievals follow. The additional stratifications shown below may have a smaller number of observations in each bin. However, they also have larger differences.

## 4.2 Sensitivity to the viewing angle

One test of the utility of AMDAR reports as a validation standard regards the recurring question of the degree to which satellite retrieval performance varies as a function of scan angle. IASI scans up to 48 degrees on either side of nadir, with scans further away from nadir having a longer geometric path through the atmosphere and larger spatial footprint. Fig. 3 explores the impact that these factors might have on the accuracy of the profile retrievals by evaluating IASI performance as a function of scan angle. All observations were sorted into bins with 10-degree increments, except for the highest bin which includes the

maximum zenith angle of approximately 58.5 degrees; the discrepancy between the maximum scan and zenith angles is a result of the curvature of the Earth. Darker blue colours in Fig. 3 represent profiles from viewing angles closer to nadir while lighter green colours indicate more oblique views. It is evident that the cool temperature bias near the surface and tropopause (Fig. 3 (left)) generally becomes colder with increasing scan angle, increasing from about 0.1 K to 0.4 K; mean differences are smaller in the middle troposphere with very little angle dependence between 60- and 700 hPa. The uncertainties also have a small, but

discernible, dependence on scan angle, with the more vertically-pointing views tending to have smaller random error than the more slanted views.  The spread in the uncertainties is largest above 400 hPa, where they differ by approximately 0.13 K.

Similar behaviours can be seen with respect to SH differences, which remain underestimated throughout the depth of the troposphere.   In the lower half of the troposphere, where the largest amount of moisture is concentrated (Figure 3 middle),

results show slightly worsened biases and uncertainty at the higher zenith angles. Overall, the differences caused by viewing angle changes in IASI humidity profiles remains consistent within 0.1 to 0.2 g kg$^{-1}$, which is small compared to the uncertainty budget. The profiles of derived RH bias and uncertainty (Fig. 3 (right)) do not show as clear a dependency on scan angle as temperature and mixing ratio. The largest dispersion is observed for the widest angles but stays well within 2 % RH on average. Again, this is likely due to the offsetting biases of the temperature and water vapor observations.



## 4.3 Differences between day and night retrievals

With a clear demarcation between daytime and night-time nodes, the Metop satellites view CONUS under two very different solar conditions. While CONUS radiosonde launches tend to be near dawn and sunset, the AMDAR observations have a much broader distribution which enables direct evaluation of both daytime and night-time observations. Fig. 4 explores the differences in IASI performance as a function of day versus night. Overall, the day and night statistics are individually quite close to the bulk statistics presented previously, with both nodes showing persistent dry biases throughout the analyzed depth, but slight differences exist. Near 1000 hPa, the night-time observations show a relatively small warm bias of approximately 0.2 K while the daytime observations have a slight cool bias of less than 0.1 K; this would only be noticeable for coastal locations as most other airports are located at lower pressures. The differences between the temperature biases lessens with increasing height up to 850 hPa, at which point they are effectively the same throughout the rest of the analyzed depth. The differences in observation uncertainty between day and night are very small.

The differences in SH are also small, but slightly more pronounced. Differences in biases are most prevalent below 750 hPa, with daytime retrievals being as much as of up to 0.15g kg$^{-1}$ too dry. While the night-time retrievals are slightly less biased than the daytime ones throughout the lowest 250 hPa of the atmosphere. Subtle differences in the derived RH bias are most prevalent between 750 and 950 hPa with day-night variations of 1 to 2 % RH at the maximum. As seen before, there is little difference in the temperature bias but more noticeable differences in the SH bias that can lead to a stronger dry bias during the day than at night at those levels. Outside of that range, differences in the RH biases between day and night are small as the temperature and water vapor biases either offset or are too small to have an impact. More significant day-night differences are observed in the variability of the SH measurements. As measured by the standard deviation, daytime retrievals are consistently more precise than night retrievals by 0.1 to 0.3 g kg$^{-1}$ throughout the troposphere. This is also reflected in the RH statistics, where the derived daytime retrievals show up to 3 % less variability from AMDAR reports throughout the troposphere than at night. The improved water vapor sensing performance during daytime could be explained by the fact that surfaces are then warmer than at night, resulting in stronger thermal contrasts, which are more favourable for atmospheric sounding.

## 4.4 Differences as a function of season

The availability of AMDAR reports across the full CONUS provides a unique opportunity to validate LEO products derived from multiple overpasses throughout the year. The year-long evaluation dataset also affords the opportunity to evaluate whether the aircraft reports can be used to determine how IASI system performance varies as a function of season. For these tests, the matchup data from 2017 were binned into the standard seasons according to the month in which they were taken: winter (January, February, and December); spring (March, April, May); summer (June, July, and August); and autumn (September, October, and November). Since the data presented here are limited to calendar year 2017, the winter season is discontinuous. Results are shown in Fig. 5.



Compared to the day/night differences, the seasonal analysis indicates greater contrasts in performance, with the differences in seasonal temperature biases being larger than 0.5 K at most levels of the atmosphere; above 300 hPa the bias can change by
upwards of 1 K between seasons. However, there appears to be little relationship between overall environmental temperature and the magnitude of the bias; for example, the winter and summer bias profiles cross repeatedly throughout the analyzed depth. The spread in the profiles of temperature standard deviation is also larger seasonally than diurnally, with higher precision being reached at summertime.

While there is a large spread in SH uncertainties, the uncertainties themselves largely correspond with the absolute amount of water vapor typically present in the atmosphere in each time and pressure level. Larger uncertainties are present nearer the surface than at higher altitudes, while substantially larger uncertainties are found during the summer than during the winter when the absolute water-vapor content is significantly lower. It is worth noting that the seasonal spread in the SH uncertainty only shows a small change with height, even as the magnitude of the uncertainties themselves trend downward significantly
with height. The corresponding statistics in RH profiles show little variability in bias. The most noticeable differences in agreement between the AMDAR and IASI observations occur between the surface and 700 hPa, where it can be as much as 5 %. The greatest agreement again occurs during the summer, a time when warmer summer surfaces could explain more favourable thermal contrasts for sounding.

## 4.5 Differences as a function of environmental characteristics

While it is natural to focus on retrievals as a function of pressure (and, by extension, altitude), this is not the only way to assess the performance of the IASI Level 2 products. An unanswered question remains: how well can AMDAR observations be used to determine how retrievals perform as a function of the value of the quantity being retrieved? For example, the vertical profile of standard deviation of temperature differences largely follows the vertical profile of temperature itself: highest at the surface and above the tropopause, and lowest in the middle. Are the observed differences therefore a function of pressure and altitude,
or are they actually a function of the underlying temperature? It is therefore appropriate to investigate the performance of the IASI Level 2 retrievals at various temperatures and humidity thresholds. As with other comparisons presented here, this increased level of investigation requires a more expansive dataset than can be provided by using standard radiosondes.

Such relationships are examined in greater detail in Fig. 6. The AMDAR minus IASI differences are plotted as a function of the AMDAR value used in those calculations. Each panel illustrates the median difference (dots) as well as the
interquartile range (lines) for the differences. As can be seen in Fig. 6 (top), there is a clearly identifiable trend toward increasing warm bias as temperatures drop below 220 K. Such temperatures are typically only found at near-tropopause levels, which shows consistency with the pressure-based plots discussed earlier, and only 12.1 % of the observations in this dataset are that cold. The majority of the observations reflect the slight cool bias previously observed. Between 220 K and 280 K (60.7% of the dataset), the magnitude of the bias is consistent. It is in this range, especially between 230 K and 260 K, that the



magnitude of the interquartile range is at its smallest. Between 280 K and 300 K (25.7 % of the dataset), the cool bias gets colder with increasing temperature to approximately -1.0 K at 300 K. At temperatures warmer than 300 K (the remaining 1.5 % of the dataset) the bias becomes slightly less cold with increasing temperature.

The specific humidity is captured in bins of 1 g kg$^{-1}$ width centred on integer values (Fig. 6 middle). There is a clear trend of
underestimating moisture content as SH increases. Where SH values are small, the absolute difference between the IASI and WVSS-II observations is also small, hence the lack of a discernible interquartile range associated with the driest environments even though the number of observation points is largest. The relative error may be large, however. By contrast, for environments with 20 g kg$^{-1}$ of SH the bias is approximately -3.3 g kg$^{-1}$. Most of the intercomparisons are at low values of SH, as the median SH in the intercomparison dataset is only 0.78 g kg$^{-1}$. While the width of the interquartile range is mostly constant
over a large range of specific humidities as it varies by less than 0.5 g kg$^{-1}$ between 6 g kg$^{-1}$ and 12 g kg$^{-1}$, that range encompasses only 15 % of all observations. At high moisture levels, the range of differences is small implying that IASI agrees well with AMDAR in high moisture environments where accurate assessment of moisture content could have a high impact on operational forecasting and NWP. RH (Fig. 6 bottom) is similar to the SH in that it has a trend towards increasingly large amplification of dry bias as RH approaches 100 %, but unlike SH this cannot solely be ascribed to small differences in water
vapor content. Since it is possible for a given RH value to occur at effectively any pressure or temperature, this implies that the trend towards IASI retrievals underestimating higher water vapor content is not an artifact of altitude or absolute vapor content but instead is a legitimate issue that may need further addressing.

### 4.6 Differences as a function of geographical location

As mentioned before, one of the key advantages to using the airborne dataset to evaluate satellite observations is that it is not
as spatially limited as radiosonde datasets are. While AMDAR observations at lower altitudes tend to be clustered near population centres due to the presence of major airports, at higher altitudes the entire CONUS is blanketed by airborne observations as commercial aircraft cruise between airports. This allows for the evaluation of what, if any, spatial dependency the IASI profiles may have. Fig. 7 depicts such an analysis. In this case, all intercomparisons from the 300 hPa pressure level and higher were binned according to their geographic location into 1 deg latitude by 1 deg longitude bins; this height was
chosen because the spatial extent of airborne observations is much more continuous at these cruising levels than it is closer to the surface. The mean and standard deviation of the differences in each bin were calculated and then plotted onto maps. The overarching message is that regardless of the quantity being measured, there is little systemic spatial variation, neither geographical nor in continental versus maritime observations. The bias or standard deviation of the differences are mostly uniform across the analyzed region, with the same cool and dry biases observed above 300 hPa in the vertical profile analysis.
Much of the noise that exists in these maps can be attributed to the relative differences in the number of intercomparisons in each bin. As Fig. 1 showed, there are more temperature observations than moisture ones, more observations over land than over the ocean, and more observations in the middle of CONUS than along the Canadian or Mexican borders. In Regions and





parameters with fewer observations exhibit a greater degree of variability from one bin to the next, visible as checkerboarding in the figures, than those with more observations.

## 5. Summary and conclusions


In the present work, the value of the routine observations of atmospheric conditions observed by commercial aircraft during the course of their regularly-scheduled flights for the evaluation of satellite-based observations was demonstrated. While this work used these observations to analyze the performance of IASI Level 2 temperature and moisture profiles as processed by EUMETSAT, a similar analysis could be conducted on any thermodynamic profiling satellite or retrieval system, including

the retrievals produced for various satellites by the NOAA Unique Combined Atmospheric Processing System (NUCAPS) or for existing or forthcoming hyperspectral geostationary sounders . Furthermore, separate analyses could be performed for microwave-only and infrared-only retrievals. When compared to radiosondes, the use of combined AMDAR and WVSS-II reports throughout the troposphere and lower stratosphere greatly increase the spatial and temporal coverage and density over the CONUS. Furthermore, as radiosondes are limited to standard global synoptic observing times while low-earth orbiting

profiling satellites are typically sun-synchronous, many regions have significant mismatches between satellite overpasses and radiosonde launches. Airborne observations have no such limitation and therefore constitute a very valuable reference for the monitoring of polar satellite products in ways not previously possible. As a result, evaluation of IASI performance over regions like the western CONUS are now possible.

Results showed strong consistency between the IASI sounders on Metop-A and Metop-B and aircraft reports agreed well with more limited, traditional radiosonde intercomparison available elsewhere.  The more robust AMDAR intercomparisons also revealed good consistency in retrieval bias and uncertainty across different regions of CONUS. The least amount of uncertainty for IASI temperature retrievals was generally found in the middle troposphere, during summertime days, and for near-nadir-pointing scans. SH retrievals showed the least absolute uncertainty where the least amount of water vapor was present, while

RH uncertainties generally behaved as temperature uncertainties did, though there was less dependence on satellite zenith angle.

It is worth noting that the AMDAR and WVSS-II observing systems are designed for operational meteorology. In the era of rapidly cycling forecast models, the airborne data need to be downlinked to the surface, quality controlled, and distributed to

partnering weather services for assimilation within minutes of the reporting time. This means that the airborne observations are well-suited towards near-real time monitoring of satellite performance and could be integrated into the operational workflow of satellite data processing centres.



While this initial work focused on CONUS due to the high density of airborne temperature and moisture observations, it could be expanded to any region where a significant number of observations from AMDAR-participating airlines exist. This includes Europe, eastern Asia, the Caribbean, and (at cruise level) the north Atlantic. Further studies of the performance of only temperature data provided by LEO retrievals could be expanded into these areas without added instrumentation. Future work will be to expand the scope of the analysis to these regions, to introduce new classifications like performance over different surface types, meteorological situations, and to evaluate how different matching and quality control criteria impact the magnitude of the biases and uncertainties. The paucity of radiosonde observations worldwide means that such investigations would be difficult using traditional evaluation methods, but the greater spatial and temporal coverage of airborne observations would facilitate these novel evaluations.

**Data Availability**

AMDAR and WVSS-II observations were obtained from the NOAA Meteorological Assimilation Data Ingest System (MADIS, madis.ncep.noaa.gov). IASI Level 2 products are available at the EUMETSAT Data Store (data.eumetsat.int).

**Acknowledgements**

This research was performed in collaboration with the EUMETSAT Remote Sensing and Products Division under the auspices of the EUMETSAT Visiting Scientist Program. Additional support was provided by the NASA Decadal Survey Incubation Program under federal grant 80NSSC22K1100 and through the United States National Weather Service Office of Observations through federal grant NA15NES4320001.

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



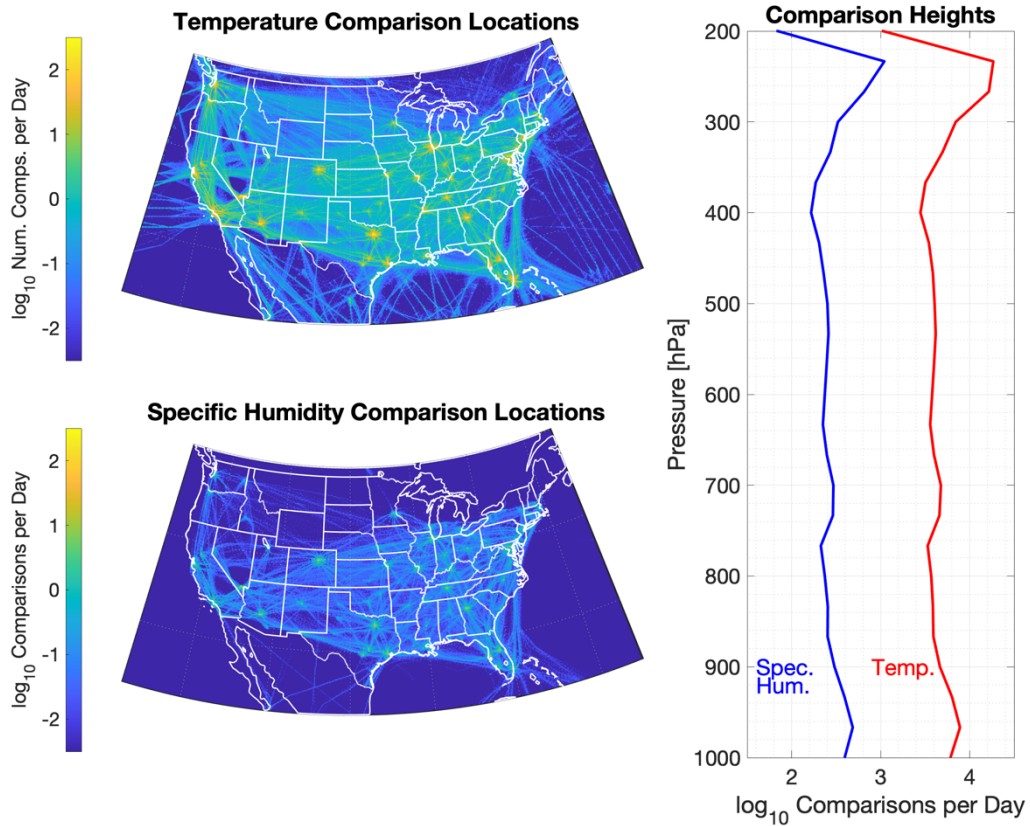

**Figure 1: Maps of the spatial distribution of the locations of IASI-AMDAR temperature (top left) and moisture intercomparisons (bottom left) over CONUS for all of 2019 plotted as the base 10 logarithm of the number of observations per 0.1-degree latitude by 0.1-degree longitude box per day. Colour scales are identical between the two maps to facilitate intercomparison. The vertical distribution of the number of intercomparisons as a function of pressure is also shown (right) for both temperature (red) and water vapor (blue). Data were binned into three bins per every 100 hPa, and the base 10 logarithm of the number of intercomparisons per bin per day is plotted.**






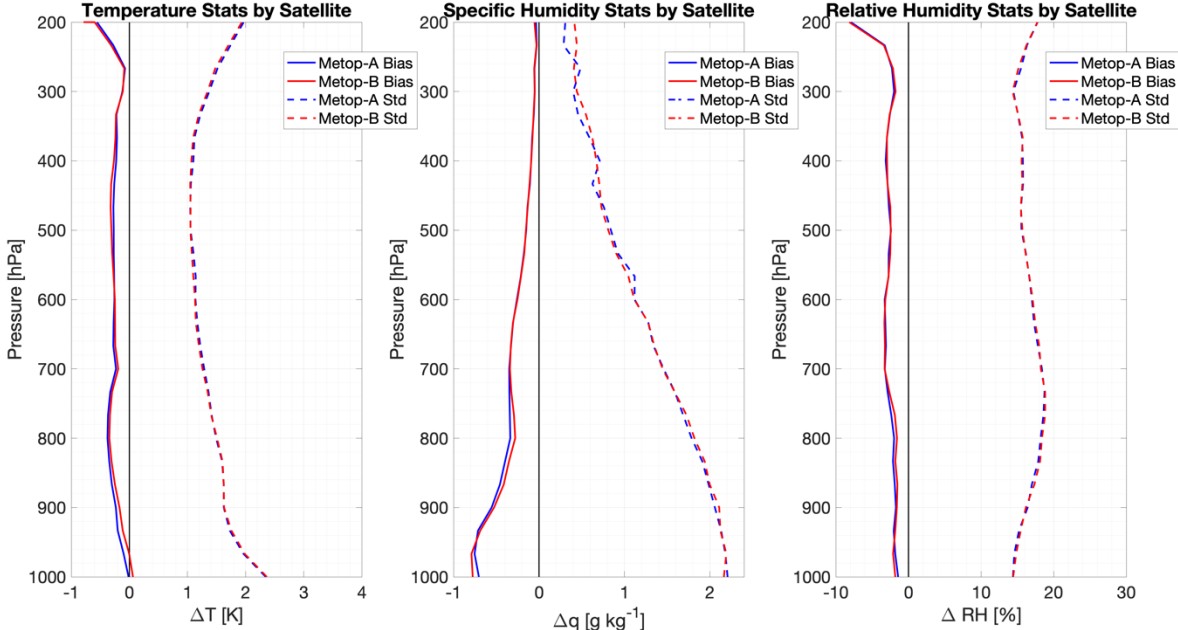

**Figure 2. Vertical profiles of the bias (solid) and standard deviation (dashed) of the AMDAR-minus-IASI Level 2 profile retrieval differences for temperature (left, in K), SH (center, in g kg⁻¹) and RH (right, in percent). Retrievals from Metop-A are shown in blue while Metop-B retrievals are shown in red. The relative number of observations in each vertical bin is consistent with the vertical profile shown in Fig. 1.**





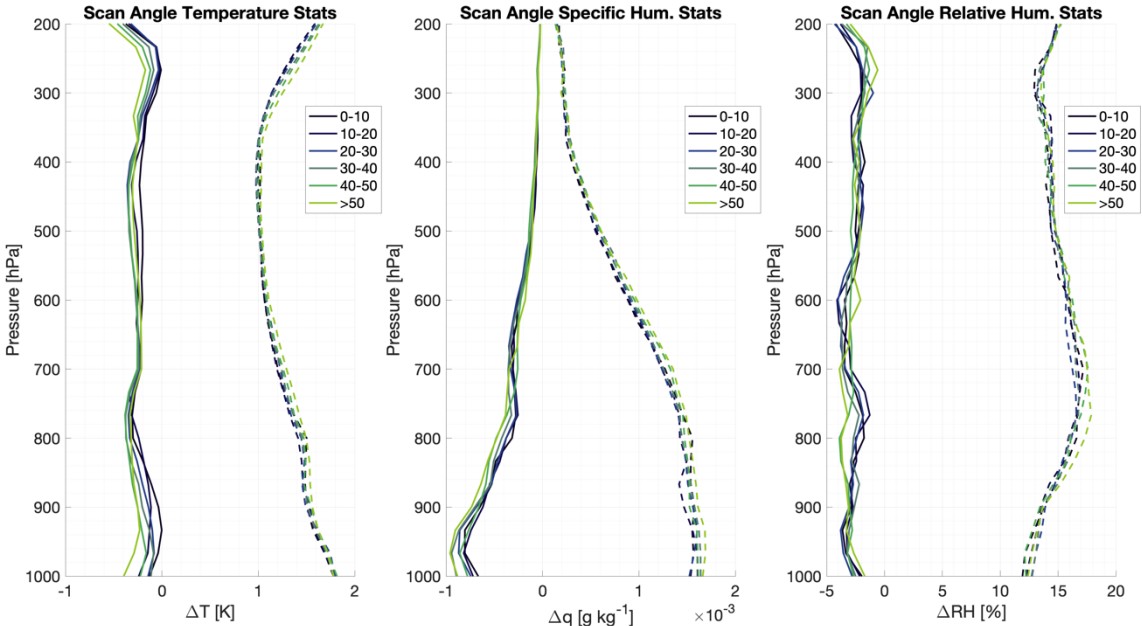

**Figure 3: As in Fig. 2, but for different satellite zenith angles. Darker blue lines represent more nadir-pointing views while lighter green lines represent more oblique angles.**






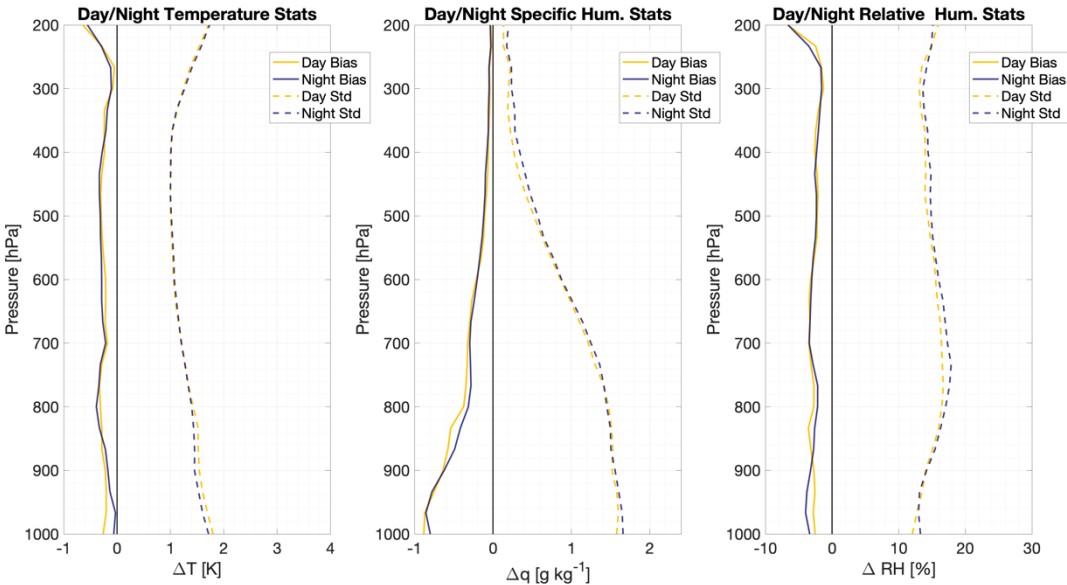

Figure 4: As in Fig. 2, but for the daytime (descending) nodes (yellow) and nighttime (ascending) nodes (blue).




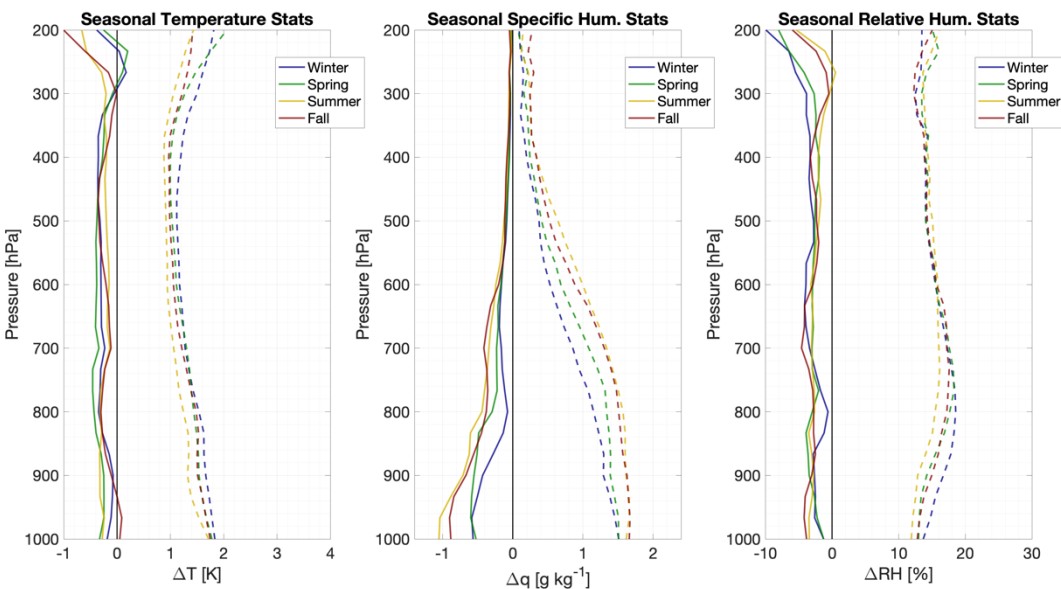

**Figure 5: As in Fig. 2, but for the four seasons of winter (blue), spring (green), summer (yellow), and fall (red) of 2019.**



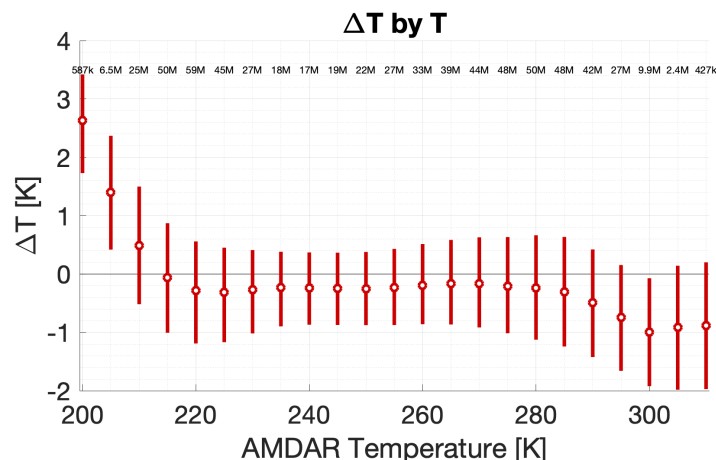

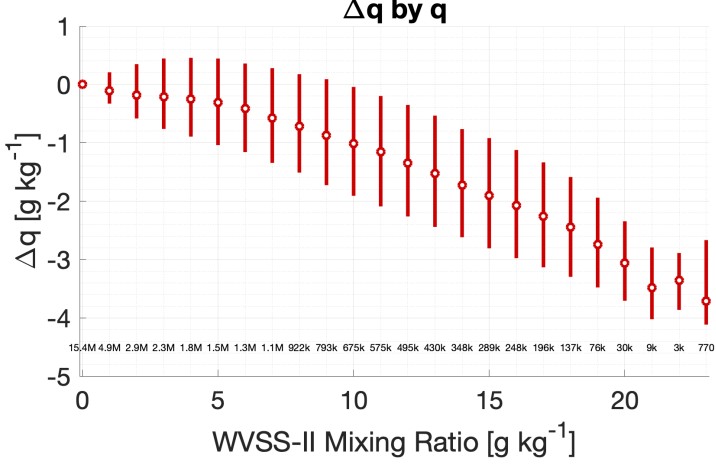

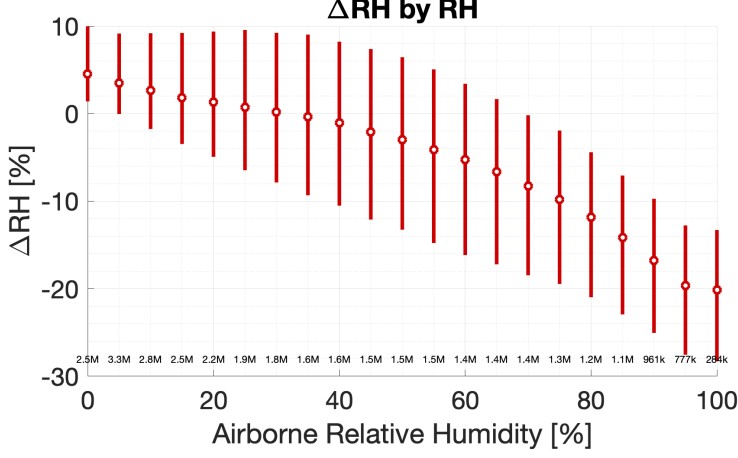



**Fig. 6: Plots of the distribution of observed IASI minus AMDAR differences for temperature (top, in K), SH (middle, in g kg$^{-1}$), and RH (bottom, in percent) as a function of the AMDAR-observed value for that quantity. Bins are 5 K, 1 g kg$^{-1}$, and 5 % respectively. Lines extend from the 25$^{th}$ to the 75$^{th}$ percentile in each bin, while the dots indicate the median for that bin. The number of observations in each bin is also displayed; larger numbers are rounded to the nearest thousand or 0.1 million as appropriate.**






**Fig. 7: Spatial variability of bias (left column) and standard deviations (right column) for temperature (top row, in K), SH (middle row, in g kg$^{-1}$), and RH (bottom row, in percent) for all IASI – AMDAR/WVSS-II differences for observations at or above the 300 hPa pressure level. Intercomparisons were spatially grouped into 1 deg latitude by 1 deg longitude bins. Dark gray areas represent regions with no observations.**
