# Peer review of "On the Use of Routine Airborne Observations for Evaluation and Monitoring of Satellite Observations of Thermodynamic Profiles"

_EGUsphere, 2023_

## Referee Comment (RC3)

General:

In my opinion, the paper is well written and demonstrates the advantages of having (and using) civil aircraft-based tropospheric observations for evaluating satellite data products. In the following I have major comments (eg a suggestion for performing further analyses) and minor comments (minor clarification in the text of the manuscript).

Major comments:

1: the authors calculate mean and discuss small details of the differences in the mean values (or biases). My question here is, if these differences in the baises are statistically significant. Maybe the authors could add the standard errors of the mean and mention in the discussion to what extend their obserserved differences (eg seasonal differences in the biases) are really significant. Mybe they are highly significant, because of the large number of independent observation that are compared, but this significance is not mentioned in the text.

2: Concerning specific humidity, the authors mention that the uncertainties (and differences in the bias they observe) are the larger the higher the specific humidity values. I think this is well understandable, and it might be useful to analyse also the relative uncertainty and biases of specific humidity. Maybe then other details become visible.

3: Maybe my most important comment, but at the same time a comment whose consideration would require most work: the authors analyse dependencies on the bias/quality of the satellite data with respect to the instrument/observing geometry (Figs. 2 + 3) and radiative or atmospheric conditions (Fig. 4 + 5). Given the large number of very good collocation they have, I was wondering whether the analyses on performance for different atmospheric condition could be further detailed. Personally, I think it could be interesting to investigate the satellite data performance for different categories of vertical layering. How is the performance for a well mixed vertical troposphere (relatively weak tropospheric humidity decrease with altitude, also relatively low temperature gradient) if compared to a highly stratified layering (exceptional humid boundary layer and at the same time a dry free troposphere, large temperature gradients). I think, this could give interesting insight into the data reliability; however, I also understand that the authors in this paper maybe first want to show the general advantages of using the AMDAR and WVSS-II data instead of only using the operational radiosonde data.

Minor comments:

Page 3, line 61: the authors might also think in adding other civil aircraft atmospheric observations like those from IAGOS.

Page 2, line 62: better write here AMDAR and WVSS-II, because you only mention at page 5 that you use AMDAR for both datasets.

Page 2, line 63 - page 3, line 67: please check, there seems to be repeating information.

Page 6, line 169-172: maybe mention that the IASI vertical resolution of the respective temperature and humidity product is good enough to use the IASI data without information on the vertical resolution (remote sensing averaging kernels).

Fig. 6: also related to my major comment 2: It seems that even the specific humidity relative error increases with specific humidity. At 10 g/kg, it is -1/10=-10%, and at 20 g/kg, it is -3/20=-15%. Maybe this could also be discussed in some way or the other.

Fig. 7: bias much smaller than std. What about the standard error of the mean? Is it much smaller than the std? So are these bias patterns significant? I have the same questions on significance of the bais differences for Figs. 2-5 (see my major comment 1).

---

## Author Comment (AC1)

**Reviewer #1**

I thought this paper was very well written and an excellent research topic. The paper is clear, consise and well thought out. This paper points out the continued need for aircraft based observations even with thie abundence of satellite observations which is an important concept for the global weather community to recognize.

We thank you for the strong words of support for our work.

A few minor comments:

1. Line 224 spelling issue at the start of the sentence "Iit"

Thank you for catching this typo. It has been fixed.

2. Figure 1, 5 both say 2019but paper states 2017

We have updated the captions with the correct dates.  Thank you.

3.Line 149. WVSS-II sensor, a tunable diode laser absorption spectrometer.

We have updated the language here.

Wondering if any of the biases with WVSS-II could be off due to the accuracy range of the sensor which is  +/- 50ppmv or +/- 5% of reading, whichever is greater.

We did consider this possibility in the text, and in Section 4.1 we noted that the sign of the bias and magnitude of the uncertainty is consistent with prior WVSS-II/radiosonde validation studies (Wagner et al. 2021, Williams et al. 2021, see paper for full references).

---

## Author Comment (AC2)

**Reviewer #2:**

This paper uses routine aircraft-based temperature and humidity measurements from the AMDAR and WVSS-II programs, respectively, to evaluate the operational thermodynamic retrievals from the IASI instruments aboard Metop-A and Metop-B. They find that the retrievals from the two satellites have almost identical biases relative to the aircraft obs. Furthermore, the evaluation of the satellite retrievals by day/night, viewing angle off nadir, and season show very small relative differences.

I find this paper very interesting, and very relevant for this journal. It is well written, the figures are clear and informative, and the conclusions generally well supported by the writing. I believe that this manuscript only needs a small amount of work to make it suitable for publication in EGUsphere. I have one "major" (really not too major) and several minor comments.

We thank the reviewer for the positive comments and their support of this manuscript for eventual publication.

Major comment:

Does the IASI team independently retrieve RH and SH (i.e., retrieve the two variables separately)? It seems that they must, because otherwise the mean biases don't make sense to me. I have two examples that I'll use to illustrate my confusion:

The IASI Level 2 algorithm retrieves SH. The values for RH used in this study are postprocessed from the retrieved T and SH by first converting the SH to vapor pressure and then obtaining the saturation vapor pressure via the Bolton formula:

$$e_s(T) = 611.2 \ \exp\left(\frac{17.67 \ T_c}{T_c + 243.5}\right)$$

where $e_s$ is the vapor pressure in pascals and $T_c$ is the temperature in Celsius (not Kelvin). This conversion was carried out by the team and is not a part of the routine IASI Level 2 output. We have added some text on this point to the manuscript to aid readers.

Fig 6: at 205 K, the bias in SH is about 0 (or slightly negative) and the bias in T is positive. For a given SH, a positive error in T would result in a decrease in RH, and similarly, decreasing SH should result in a decrease in RH. But for the temperature bin, the RH bias is +3 RH%. I know we are looking at means here, but the tails of the distribution (for that T) are even worse. This is inconsistent.

The three panels in Fig. 6 are independent. The top panel is showing bias in temperature as a function of temperature.  The middle panel displays bias in SH as a function of SH, not temperature, and the RH panel shows bias as a function of RH, not as a function of the other variables. This was done in order to separate the analysis of the variables from the confounding impacts of altitude and pressure which are implicitly present in Figs. 2–5. Furthermore, the temperature statistics represent a much larger and more diverse dataset than the SH and RH observations do, as indicated in the maps in Fig 1. Therefore, quantitative comparisons between the various panels are not possible.

In Fig 2 at 1000 mb, the bias in T is zero and the bias in q is -0.8 – which seems to lead to a bias in RH of about 2 %RH. I did some back-of-the-envelope calculations, and for a range of T (from 10 to 35 C), and starting from two points (roughly RH=80% and RH=40%) and assuming that the aircraft data were about 0.8 g/kg drier than IASI (to give the bias shown in Fig 2), the resulting difference in RH ranges from -10 %rh to -2% RH.  Since there is a year of observations from 2017 in this figure, I think that the mean RH bias should be much later than the 2% shown in Fig 2 if the q and RH retrievals from IASI are consistent with each other (I think the mean RH bias should be closer to -5 %rh).  (see the attached figure on the next page)

In the example at 1000 hPa, an unbiased temperature and a negatively-biased SH should lead to a negatively-biased RH, which is what is observed.  However, the relationship between temperature and relative humidity is nonlinear and asymmetrical. While the random temperature uncertainties may average to zero bias, those uncertainties may have a non-zero impact on the RH bias.

For example, let's assume conditions of 20 C, 11.7 g/kg, and 1000 hPa. This works out to a RH of 80.0% according to the Bolton formula described above; this looks to correspond well with the figure you included with your review. We did a quick monte carlo analysis in which we held the SH constant but perturbed the T by random values chosen from a normal distribution with a mean of 0 and a standard deviation of 2.4 (the uncertainty calculated at 1000 hPa in Fig. 2).  After a million calculations, the mean RH was 81.0%. Despite unbiased temperatures going into the calculation and no change to the specific humidity, the net result was an RH that was biased moist by an entire percentage point. Such values partially offset the dry bias from the analysis you included with your review. The difference in scope between the airborne temperature and moisture datasets is an additional hindrance to these quantitative comparisons.

Anyway, I think the authors need to be more explicit about the consistency between the IASI retrieval of q vs RH (or if one is derived from the other), and if they are independently retrieved, to spend more time discussing the implications of this.

We do appreciate this concern, and we have added discussion on this issue to the text. We thank you for encouraging us to be more explicit and precise in our writing.

Minor comments:

We thank you for each of these suggestions, which significantly improve the clarity and readability of our work.

- Line 41: twice-daily revisits by a single satellite – this should be clarified

Clarified.

- Line 93: "first constituted with of the order of 10^8 of real IASI" – this is very awkward, and should be rewritten

Changed to "...a training base is constructed with over 100 million real IASI and AMSU/MHS observations collocated with model reanalysis data..."

- Line 149: a laser diode hygrometer

We have changed this to slightly different language based on the suggestion of Reviewer #2.

- Line 177: there is also significant displacement (and perhaps even more) for descents. The way this was written suggests that only ascents (not descents) were analyzed here, which I don't think was true. Please update

You are correct. The sentence has been updated.

- Line 273: does the shape of the vertical profile matter to the IASI retrieval? For example, in the daytime, the water vapor specific humidity is often pretty constant in the convective boundary layer, but that isn't true at night. This should at least be added to the text as a possible explanation for the small day/night differences in the bias

We have included the potential for shape to influence the retrieval accuracy.

- Caption of Fig 1 and Fig 5: you state "2019" when I am pretty sure you mean 2017

You are correct, and the captions have been updated.

- Caption of Fig 2: you state "AMDAR-minus-IASI" when all of the other results are the reverse. I believe this is a typo, based upon the results shown later

This, too, was a mistake on our part, and we thank you for catching that.

---

## Author Comment (AC3)

**Reviewer #3, Matthias Schneider**

We received this reviewer's comments during the pre-review phase that determined if this paper should be fully reviewed for AMT. We addressed the reviewer's valuable comments at that stage and wrote a point-by-point response describing how we implemented those comments between the AMTD and AMT submissions. We apologize that it was not clear to the reviewer if their points had been addressed in the publicly-posted version of the manuscript. Below, we are reproducing our response to the reviewer from that stage.

General:

In my opinion, the paper is well written and demonstrates the advantages of having (and using) civil aircraft-based tropospheric observations for evaluating satellite data products. In the following I have major comments (eg a suggestion for performing further analyses) and minor comments (minor clarification in the text of the manuscript).

We thank the reviewer for the time spent on evaluating this work and determining its suitability for further review for AMT.

Major comments:

1: the authors calculate mean and discuss small details of the differences in the mean values (or biases). My question here is, if these differences in the baises are statistically significant. Maybe the authors could add the standard errors of the mean and mention in the discussion to what extend their obserserved differences (eg seasonal differences in the biases) are really significant. Mybe they are highly significant, because of the large number of independent observation that are compared, but this significance is not mentioned in the text.

The reviewer is correct in that the large number of observations means that even the smallest differences are statistically significant. For example, Figure 2 notes the differences between the two IASI instruments. While the difference curves have almost identical shapes, there are hundreds or thousands of observations in each bin. As a result, a two sample t-test for the difference of mean indicates that the mean temperature differences that are statistically significant at the 95% confidence interval at every height, despite the biases being different by less than 0.01 K at certain heights. While the other stratifications shown here may have smaller bins, they also feature larger differences. We have added this discussion to the paper.

2: Concerning specific humidity, the authors mention that the uncertainties (and differences in the bias they observe) are the larger the higher the specific humidity values. I think this is well understandable, and it might be useful to analyse also the relative uncertainty and biases of specific humidity. Maybe then other details become visible.

The challenge with evaluating the relative uncertainties is that many of the observations contain very small amounts of water vapor as aircraft spend most of their time at cruising altitude where absolute water vapor content is small. By far the majority of observations have a specific humidity of 1 g/kg or less, and thus most of the time the small differences are amplified when evaluating from a relative perspective as we have to divide by numbers much less than one. Therefore, we chose to focus on absolute uncertainties.

3: Maybe my most important comment, but at the same time a comment whose consideration would require most work: the authors analyse dependencies on the bias/quality of the satellite data with respect to the instrument/observing geometry (Figs. 2 + 3) and radiative or atmospheric conditions (Fig. 4 + 5). Given the large number of very good collocation they have, I was wondering whether the analyses on performance for different atmospheric condition could be further detailed. Personally, I think it could be interesting to investigate the satellite data performance for different categories of vertical layering. How is the performance for a well mixed vertical troposphere (relatively weak tropospheric humidity decrease with altitude, also relatively low temperature gradient) if compared to a highly stratified layering (exceptional humid boundary layer and at the same time a dry free troposphere, large temperature gradients). I think, this could give interesting insight into the data reliability; however, I also understand that the authors in this paper maybe first want to show the general advantages of using the AMDAR and WVSS-II data instead of only using the operational radiosonde data.

The intent of this paper is to show the general suitability of airborne observations for evaluating satellite-observed thermodynamic profiles. We share the reviewer's interest in further stratification of the data in order to evaluate the satellite performance in different observation types. In fact, we are currently conducting work that shows an increasing underestimate in convective available potential energy (CAPE) with increasing CAPE values, likely due to an increasing low level dry bias in more unstable environments. A teaser of that work is shown below. This and related analyses are beyond the scope of the current paper, the aim of which is to describe and demonstrate methodology.

[Figure]

Minor comments:

Page 3, line 61: the authors might also think in adding other civil aircraft atmospheric observations like those from IAGOS.

The focus of this paper is on the thermodynamic products from satellite and their validation. The IAGOS aircraft are much fewer in number and focus more on atmospheric composition, which we are not evaluating at this time. We have slightly modified the wording in this paragraph to stress that we are evaluating thermodynamic profiles. Overall, we feel that the inclusion of IAGOS brings more confusion than clarity to this discussion.

Page 2, line 62: better write here AMDAR and WVSS-II, because you only mention at page 5 that you use AMDAR for both datasets.

Thank you for suggesting this change which increases readability. We have made it.

Page 2, line 63 - page 3, line 67: please check, there seems to be repeating information.

That is correct, and we have edited these sentences to omit the repetition.

Page 6, line 169-172: maybe mention that the IASI vertical resolution of the respective temperature and humidity product is good enough to use the IASI data without information on the vertical resolution (remote sensing averaging kernels).

As this is a paper devoted to observational techniques, we feel that making claims about the specific attributes of the data for assimilation may be beyond the scope of what this paper is addressing. Regardless of the true vertical resolution of an instrument, the information content is still coming from a layer of the atmosphere instead of a specific height, and averaging kernels help ensure that the observations are properly distributed.

Fig. 6: also related to my major comment 2: It seems that even the specific humidity relative error increases with specific humidity. At 10 g/kg, it is -1/10=-10%, and at 20 g/kg, it is -3/20=-15%. Maybe this could also be discussed in some way or the other.

When we are talking about relative error in this sense, we are referring to the fact that at high altitudes, the absolute values in observed water vapor are very small.  Therefore even small absolute differences can manifest themselves as large relative differences when the baseline value is much less than 1 g/kg.

Fig. 7: bias much smaller than std. What about the standard error of the mean? Is it much smaller than the std? So are these bias patterns significant? I have the same questions on significance of the bais differences for Figs. 2-5 (see my major comment 1).

Since the standard error of the mean is simply the standard deviation divided by the square root of the number of the observations, it goes to zero with an increasing number of observations. For the bins with a non-zero number of observations, the median number of temperature observations in a bin is approximately 2800 and some bins have well over $10^5$ observations; moisture observations are roughly one order of magnitude smaller in number. As a result, SEM values for this figure are on the order of 0.05 K (0.1 g/kg) or less, much smaller than the uncertainty as represented by the standard deviation.